# Detecting Skin Reactions in Epicutaneous Patch Testing with Deep Learning: An Evaluation of Pre-Processing and Modality Performance

**DOI:** 10.3390/bioengineering10080924

**Published:** 2023-08-03

**Authors:** Ioannis A. Vezakis, George I. Lambrou, Aikaterini Kyritsi, Anna Tagka, Argyro Chatziioannou, George K. Matsopoulos

**Affiliations:** 1Biomedical Engineering Laboratory, School of Electrical & Computer Engineering, National Technical University of Athens, 9 Iroon Polytechniou St., 15780 Athens, Greece; ivezakis@biomed.ntua.gr (I.A.V.); glamprou@med.uoa.gr (G.I.L.); 2Choremeio Research Laboratory, First Department of Pediatrics, National and Kapodistrian University of Athens, 8 Thivon & Levadeias St., 11527 Athens, Greece; 3University Research Institute of Maternal and Child Health & Precision Medicine, National and Kapodistrian University of Athens, 8 Thivon & Levadeias St., 11527 Athens, Greece; 4First Department of Dermatology and Venereology, “Andreas Syggros” Hospital, National and Kapodistrian University of Athens, 5 Ionos Dragoumi St., 11621 Athens, Greece; katerinakyr18394@gmail.com (A.K.); annatagka3@gmail.com (A.T.); haciargiro.ac@gmail.com (A.C.)

**Keywords:** patch testing, contact dermatitis, allergic contact dermatitis, deep learning, machine learning, image analysis, pre-processing, skin reactions, diagnosis, automated diagnosis

## Abstract

Epicutaneous patch testing is a well-established diagnostic method for identifying substances that may cause Allergic Contact Dermatitis (ACD), a common skin condition caused by exposure to environmental allergens. While the patch test remains the gold standard for identifying allergens, it is prone to observer bias and consumes valuable human resources. Deep learning models can be employed to address this challenge. In this study, we collected a dataset of 1579 multi-modal skin images from 200 patients using the Antera 3D^®^ camera. We then investigated the feasibility of using a deep learning classifier for automating the identification of the allergens causing ACD. We propose a deep learning approach that utilizes a context-retaining pre-processing technique to improve the accuracy of the classifier. In addition, we find promise in the combination of the color image and false-color map of hemoglobin concentration to improve diagnostic accuracy. Our results showed that this approach can potentially achieve more than 86% recall and 94% specificity in identifying skin reactions, and contribute to faster and more accurate diagnosis while reducing clinician workload.

## 1. Introduction

Allergic Contact Dermatitis (ACD) is a prevalent immunological reaction of the skin related to exposure to environmental substances, affecting all age groups as well as both genders [1]. More than 5200 chemicals have been identified as allergens that can cause ACD [2]. Most commonly found on the hands and face, the symptoms manifest as eczematous dermatitis and include skin dryness, itchiness, and inflammation. Identifying the allergen(s) causing ACD is imperative for its proper management and/or treatment.

The epicutaneous patch testing method, although dated back to 1895, currently remains the gold standard for identifying the allergens that can cause ACD to a patient [3,4]. It involves the selection of an appropriate series of allergens most likely to cause ACD based on a patient’s history and the nature of their skin condition, and their re-exposure to the allergens under controlled conditions to verify the diagnosis [5]. Common allergens include metals (such as nickel), fragrances, preservatives, and rubber chemicals. A patch test unit is a specially manufactured tape fitted with sockets, called “chambers”, that contain the allergens themselves. The patch test is directly applied to the skin, typically on the upper back. It remains in place for 48 h and is then removed. After another 24–48 h, the area is examined for signs of skin reaction such as redness, swelling, or itching. The test is relatively safe, and the reactions are usually resolved within a few days. However, in rare cases, severe reactions may occur [4]. The results of the patch test help towards the identification of the specific allergens that trigger an allergic reaction to the patient, reducing the risk of future ACD episodes.

The International Contact Dermatitis Research Group (ICDRG) criteria for reading patch tests are globally acknowledged and rely on the visual inspection and palpation of the reaction morphology, including erythema, infiltrate, papules, and vesicles [6]. However, despite its effectiveness as a diagnostic tool, patch testing is subject to observer bias, as result interpretation can vary between observers. For example, inter-observer variability has been identified in the discrimination between doubtful and irritant reactions, as well as in the distinction between doubtful and weak positive reactions. To mitigate this, continuous standardization and reading training are recommended [6]. Johansen et al. also report that to ensure a uniform inter-individual patch test reading technique, continuous training is necessary, for example, in the context of “patch test courses” at scientific meetings. This further highlights the subjectivity involved in reading patch tests, as it requires a certain level of expertise and training to accurately interpret the results. Beyond observer bias, patch testing also demands valuable human resources, a challenge that can be particularly pronounced in large clinics. To address similar challenges in medical settings, researchers have proposed the use of deep learning models [7,8,9]. However, the application of deep learning to automate the detection of skin reactions in patch testing remains largely unexplored.

In this work, we aim to address these gaps by investigating the potential of a deep learning model to automate the evaluation of patch tests, as well as the impact of different pre-processing techniques and image modalities on classification accuracy. We collected a dataset of 1190 multi-modal images using the Antera 3D^®^ camera depicting skin reactions to a series of allergens, from 200 patients. Our study proposes a pre-processing technique that retains the contextual information of the images in order to increase the accuracy of the deep learning classifier. Moreover, we explore the integration of multiple image modalities captured by the camera, including false-color maps of hemoglobin concentration, texture, volumes, fine lines, and folds, to improve diagnostic accuracy.

In summary, our work makes the following contributions:We evaluate the feasibility of using a deep learning classifier to automate the identification of allergens causing ACD, reducing the reliance on visual inspection and potential observer bias.We introduce a novel, yet simple, context-retaining pre-processing technique that significantly improves the accuracy of the classifier.We explore the additional value provided by multi-modal images.

By addressing these contributions, our research has the potential to significantly advance the automated diagnosis of ACD with epidermal patch testing, enabling faster and more accurate identification of allergens while reducing clinician workload. To the best of our knowledge, there is only one other existing work in this area (Chan et al. [10]), making our study a valuable starting point for future endeavors.

The paper is structured as follows: Section 2 provides an overview of the materials and methods, including dataset collection and deep learning experimental setup. In Section 3, we present the results of our experiments, covering pre-processing scheme comparison, modality performance comparison, and investigation of combined modalities. Section 4 discusses the findings and highlights limitations. Finally, Section 5 presents the conclusions drawn from our study and outlines future research directions.

## 2. Materials and Methods

### 2.1. Dataset

The dataset used in this study includes data from 200 patients, collected in the Laboratory of Patch Testing, National Referral Centre of Occupational Dermatoses, University Hospital “Andreas Syggros,” National and Kapodistrian University of Athens, Medical School. The detailed clinical protocol is described in Section 2.1.3.

#### 2.1.1. Inclusion Criteria

All patients were admitted to the hospital with the suspicion of contact dermatitis [11,12]. In order to avoid bias due to selective inclusion, 200 patients were randomly selected and included in the dataset.

#### 2.1.2. Exclusion Criteria

Although there is no consensus for the definitive criteria of exclusion from a patch testing procedure, most studies agree that the use of corticosteroids [13], immunodepressants (such as cyclosporine and chemotherapeutics) [14], anti-inflammatory treatments [6], other chronic dermatopathies [13], and high exposure to UV [15] (as for example during the summer) are factors that might produce false-negative or false-positive results and should thus be avoided. As such, patients admitted who were under anti-inflammatory treatment, under cyclosporine treatment, chronically used corticosteroids, under chemotherapeutics treatments, as well as suffering from other chronic dermatopathies, were excluded from the present study. In addition, we excluded patients younger than 18 years and older than 65 years.

#### 2.1.3. Patch Testing and Clinical Evaluation

The European baseline series contains various categories of allergenic factors such as metals, fragrances, preservatives, rubbers, topical therapeutics, and excipients that may cause contact dermatitis [16]. In this present study, sensitization was evaluated using a battery of 30 allergens from the European baseline series, with additional series added in order to identify sensitizations and inform the national baseline of allergens. Table A1 in the appendix contains the complete list of the allergens used.

We conducted the patch testing according to the guidelines of the European Society of Contact Dermatitis [6]. We applied three 5 × 2 patches on the middle upper back, to a hairless and lesion-free skin. The optimal exposure time was set to 48 h (two days). According to the respective guidelines, a test reading should be performed twice, with the first immediately following patch removal (48 h after the first exposure), and the second 1–5 days later. In the majority of patients, the maximum reaction was observed between day 3 and day 4 following allergen application. The patch test outcome was assessed based on the morphology of lesions, with reactions classified as positive/allergic on a scale of (i) + weak, (ii) ++ strong, or (iii) +++ extreme, according to the ICDRG criteria. Everything else, including irritant reactions, was classified as negative. In the present study, we reported the sensitization results only, without referring to the relevance of the patients’ clinical image and contact allergens.

#### 2.1.4. Skin Analysis

The skin was evaluated with an Antera 3D^®^ camera (Miravex Limited, Dublin, Ireland) at baseline 0 h, 48 h and 72 h following the first exposure to allergens. The Antera 3D^®^ camera employs multi-directional illumination to obtain images from different angles, which are then analyzed by its bundled software to reconstruct the skin surface in two and three dimensions. In addition, the use of LEDs emitting different wavelengths of light enables the device to perform both spatial and spectral analysis, thus capturing the skin topography and chromophores concentration. Moreover, through the camera’s polarization filters and proprietary techniques, the images acquired are independent of lighting conditions which guarantees accuracy and result reproducibility [17,18,19]. This alleviates the need for additional color constancy pre-processing techniques which are usually essential when analyzing dermoscopic images. For example, in the past, heuristic approaches were widely utilized for image homogenization and standardization [20,21]. More recently, AI techniques, such as Generative Adversarial Networks have been employed to solve this issue [22].

With the use of the Antera 3D^®^ camera, an area of 3136 mm^2^ (56 × 56 mm) was captured at a time. As this area size proved to be too small to entirely capture the skin covered by a single patch, two captures were required instead, which correspond to Area 1 and Area 2 respectively (Figure 1).

For each individual capture, the Antera 3D^®^ software (v3.0.2) produced six different images, which we henceforth refer to as “image modalities”. These images correspond to:The original Color image, which depicts the skin as seen by the naked eye.Redness, which is a false-color map used to identify the areas with higher hemoglobin concentration.Texture, an image depicting the quantification of roughness. The Antera 3D^®^ software quantifies roughness as the vertical deviation of a real surface from its ideal form, and calculates it in arbitrary units.Fine Lines, a false-color map where the dark red color indicates the deepest features, and the white corresponds to the top of the surface.Folds, which is similar to Fine Lines, capturing the same features but on a larger scale.Volumes, which depicts features protruding or depressing from the average skin plane. The false-color mapping ranges from dark red for depressions, to dark violet for elevations.

Figure 2 depicts all six modalities for a single skin area. All images were exported using the Antera 3D^®^ software, which produced an image of resolution 480 × 480 pixels, compressed using the JPEG format.

#### 2.1.5. Image Labeling and Annotation

Following diagnosis by the clinician, the bounding boxes of all allergens to which each patient tested positive to, were traced using the Antera 3D^®^ software. This was performed using the Fine Lines modality at 48 h, where the boundaries of each chamber were visible on the skin surface. The software automatically transferred the same bounding box across all timepoints (i.e., 0 h and 72 h). To avoid severe class imbalance in the dataset, the clinician additionally traced an equal amount of bounding boxes corresponding to allergens the patient tested negative to. In total, 1579 bounding boxes were traced for each timepoint. After filtering out agents that dyed the skin, namely Paraphenylenediamine and Textile Dye Mix due to the pigmentation interfering with the hemoglobin concentration mapping, 1190 bounding boxes remained for each timepoint, with the diagnosis breakdown at 72 h being as follows: 61% (n = 722) were classified as negative or irritant reactions, 18% (n = 211) as weak reactions (+), 19% (n = 230) as strong reactions (++), and 2% (n = 27) as extreme reactions (+++).

### 2.2. Deep Learning Experimental Setup

In the present study we performed three different experiments to compare the classification accuracy of a Deep Learning model across different pre-processing techniques and image modalities. Although the ICDRG criteria are designed to classify reactions into multiple classes, we opted to deal with the binary classification problem instead, classifying between negative and positive. This choice was made to avoid severe class imbalance due to insufficient data for some classes. As discussed in Section 2.1.5, the multiclassification problem would have resulted in a single class (negative) to dominate the dataset by representing 61% of all samples, with the others being as low as only 2% in the case of extreme reactions. Even if properly adjusted to take the class imbalance into consideration, the sample size would still be small. Therefore, binary classification was the most practical choice.

In the first experiment (detailed in Section 2.2.2), we compared the difference in performance when using three different pre-processing schemes on the Redness image modality. In the second experiment (detailed in Section 2.2.3), we compared the performance achieved when using each image modality separately. In the third and final experiment (detailed in Section 2.2.4), we combined image modalities to investigate evidence of added value.

#### 2.2.1. Model Configuration

For all experiments, we used an EfficientNetB0 [23] Convolutional Neural Network (CNN), pre-trained on the ImageNet dataset. We selected this network architecture as the EfficientNet family of CNNs has been previously demonstrated to achieve state-of-the-art results on ImageNet [23] and other datasets [24], while being smaller and faster than previous models. Moreover, we opted to use the B0 (smallest) variant to avoid overfitting on our rather small—compared to ImageNet—dataset. For each experiment, we trained the network for 250 epochs, using a batch size of 8. We selected the AdamW optimizer [25], and set its learning rate to decrease with a cosine annealing rate from 3 × 10^−4^ to 1 × 10^−5^ over the training course. To increase dataset diversity and avoid the network being biased in terms of allergen position, we employed data augmentation in the form of random horizontal and vertical flipping, as well as random rotation within 20 degrees. All images were normalized across their RGB channels using the mean and standard deviation of the ImageNet dataset, and resized to 224 × 224 pixels using bilinear interpolation. Throughout our experiments, we only used images obtained at 72 h after exposure, as this was the timepoint at which the maximum reaction was generally observed.

#### 2.2.2. First Experiment: Pre-processing Scheme Comparison

Data pre-processing is a crucial first step in deep learning workflows [26]. It aims to transform the raw data into a format that is easier for a deep learning algorithm to work with, while preserving the relevant information. For example, in image recognition tasks, images may be resized to a standard size and converted to grayscale to reduce computational complexity while still retaining important features such as edges and textures [27]. Data pre-processing can have a significant impact on the accuracy and performance of a deep learning model.

Each skin image in our study depicts either four or six individual allergen application spots (Figure 1). The simplest way to classify the reaction to only one allergen at a time, is to crop or mask the image to retain only the pixels corresponding to the Region of Interest (ROI), in this case the area of an individual allergen (Figure 3B). Indeed, Chan et al. [10] also cropped their images to a single allergen in their recent study for automated detection of skin reactions in patch testing. However, it is possible that this approach of cropping or masking to a particular ROI discards valuable contextual information. For instance, the difference in pigmentation between the allergen application spot and the skin area surrounding it, is a better reaction indicator than just the absolute value at the allergen. Moreover, reactions induced at neighboring chambers may also affect the ROI, resulting in false positive results. Figure 4 provides an example of this, where the reaction at allergen A is so strong that it also spreads to allergen B. If the skin at allergen B was to be examined in isolation, a classification algorithm would be very likely to falsely predict that the manifested pigmentation is a result of a reaction to this allergen.

We propose retaining the original image for improved classification performance. To this end, instead of cropping to the ROI, we applied a special case of mask, for which we coined the term “alpha mask”. In computer graphics, alpha refers to the level of opacity or transparency of an image, with 0% alpha being completely transparent, and 100% alpha being completely opaque. An alpha mask is a grayscale image ***M*** where each pixel value corresponds to the alpha, ranging from 0 to 1. The mask is applied with the following operation:(1)$$I^{\prime }=I⊙M$$
where I is the original skin image, and ⊙ is an operator for the element-by-element matrix multiplication. We arbitrarily set all pixel values in the alpha mask to 0.5, except for those belonging to the ROI, which we set to 1. In this way, we dimmed the image brightness in all places except for the ROI, effectively highlighting the area to be classified, but still retaining contextual information (Figure 3C). In this work, we refer to this scheme as “Alpha Mask” (AM).

In addition to retaining contextual information by operating on the entire original image, one might observe that the allergens of the last row in patch area 1, are adjacent to the allergens of the first row in patch area 2 (Figure 1). However, since each area corresponds to a different image, this contextual information is lost. To address this issue, we expanded each image under examination to incorporate pixels from its adjacent area. For images corresponding to patch area 1, we cropped 75 pixels from the corresponding patch area 2 and concatenated them at the bottom of the image. For images corresponding to patch area 2, we cropped 75 pixels from area 1 instead, and concatenated them at the top of the image. We named this scheme “Contextually Enhanced Alpha Mask” (CEAM). Figure 3D depicts the results obtained using this pre-processing scheme.

The evaluation of the three pre-processing schemes involved training an EfficientNetB0 with the configuration described in Section 2.2.1 on the Color image modality. We split the dataset into training and validation sets using a 70/30 split, ensuring that there was no overlap between patients in the two sets. We then independently trained and evaluated the model three times, once for each pre-processing scheme, using the exact same data split. To convert the multiclass problem into a binary classification problem, we collapsed the labels into binary values by considering “none” as the “negative” label, and the rest as the “positive” label. For each scheme, we calculated the F1 score, accuracy, specificity, recall, and precision. The details on how these are calculated can be found in Appendix D. To determine whether there was a statistically significant difference between the three pre-processing schemes, we applied McNemar’s test to the validation results. Appendix E provides a detailed explanation of how *p*-values are calculated using the McNemar test.

#### 2.2.3. Second Experiment: Modality Performance Comparison

In the second experiment, we focused on comparing the binary classification performance of EfficientNetB0 on each modality individually. To prepare the images for the experiment, we used the CEAM pre-processing scheme. To obtain a more accurate estimate of the performance metrics and further assess stability across different datasets, we trained the model using 5-fold cross-validation. *K*-fold cross-validation is a technique that divides the data into multiple non-overlapping sets, trains the classifier on K−1 sets and evaluates it on the one remaining set. By repeating this process *K* times, a performance estimate is obtained across all available data. However, we did not perform any statistical tests on the cross-validated results because all tests expect independent and identically distributed (IID) samples. This is not the case in *K*-fold cross-validation, as the training sets overlap between folds. In particular, each pair of training sets shares $1-\frac{2}{K}$ of the data. Therefore, “this overlap prevents the statistical test from obtaining a good estimate of the amount of variation” [28]. While the 5 × 2 cross-validation test has been suggested as a good alternative [28], this involves five 2-fold cross-validations and is sub-optimal for getting reliable performance estimates on small datasets due to (i) the training set being reduced more than necessary, (ii) the training data likely being significantly different from the validation data. Therefore, we opted for McNemar’s test to evaluate statistical significance while using 5-fold cross-validation to assess model performance. To perform McNemar’s test we used the 70/30 training/validation split as described in Section 2.2.2. We then trained and evaluated the model on each modality separately to determine statistical significance. We additionally evaluated the performance of the model on each individual allergen over the 5-folds by calculating the percentage of false predictions. Using this data we computed the Z-score for each allergen to identify any outliers whose data point was three standard deviations away from the mean.

#### 2.2.4. Third Experiment: Combined Modality Investigation

For the third experiment, we investigated whether combining modalities could improve overall classification performance.

Firstly, to provide an estimate of feature redundancy, we computed the Pearson correlation coefficient between the features of different modalities for the same images. To extract these features, we re-used the trained models from Section 2.2.3, and extracted the outputs from the layer preceding the final fully connected layer for each image in the validation set. To explain this further, in CNNs the final fully connected layer is the classifier. The layers preceding the classifier learn the optimal features to be used in the final classification. Therefore, by omitting the last layer, the output of the network becomes the learned features instead of its prediction. Following feature extraction, for each patient’s individual allergen images, we took the extracted features from two modalities at a time, treating them as separate variables. The Pearson correlation coefficient was calculated between these variables, giving us a measure of how linearly related the features of the two modalities are. This process was repeated for every pair of modalities for each image. To summarize the results we calculated the mean and standard deviation of the absolute correlations for each modality pair.

We then investigated whether the features resulting from each modality are complementary to each other, i.e., whether their combination can provide additional discriminative performance. To this end, like before, we combined the learned features from different modalities in pairs of two and trained a Support Vector Machine (SVM) with a Radial Basis Function (RBF) kernel on the binary classification problem. We heuristically set the C parameter to 3.5. Since the validation set consisting of 30% of the data is too small for doing an additional 70/30 split on it, we opted for cross-validation instead. In order to determine the statistical significance of our results, we followed the combined 5 × 2 cross-validation *F*-test described by Alpaydín [29] (Appendix F).

## 3. Results

### 3.1. First Experiment: Pre-Processing Scheme Comparison

Table 1 presents the F1 score, accuracy, specificity, recall and precision achieved for each pre-processing scheme. Figure 5 depicts a bar chart with the same metrics for easier visual comparison.

The results show that the highest F1 score, accuracy, and recall, was achieved with the CEAM pre-processing scheme. Conversely, the highest specificity and precision was achieved with the AM pre-processing scheme. The Crop scheme performed significantly worse across all metrics. The McNemar test (Table 2) confirmed that there is a statistically significant difference (p<0.05) between both the CEAM vs. Cropped and AM vs. Cropped pre-processing schemes. However, the same does not hold for CEAM vs. AM as the *p*-value of 0.281 does not allow us to safely assume that one scheme prevails over the other.

### 3.2. Second Experiment: Modality Performance Comparison

Table 3 shows the mean value ± standard deviation of the *F*1 score, accuracy, specificity, recall, and precision achieved for each modality over the 5-fold cross-validation. Table A2 in the appendix depicts the confusion matrices for each modality combined over all 5 folds through summation. Overall, the results indicate that the Redness modality outperforms all other modalities in terms of the mean value across all metrics. The Color modality also achieves high performance, and is found to be similar to Redness. All the other modalities resulted in significantly lower performance. Figure 6 provides a visual representation of the results.

We employed the same methodology as in Section 3.1 to identify statistically significant differences between the performance achieved when using each image modality. As shown in Table 4, we could not find a statistically significant difference between the Color and Redness image modalities, as well as Texture and Volumes, Texture and Folds, Texture and Fine Lines, Volumes and Fine Lines, Folds and Fine Lines (p≥0.05).

In order to provide a visual representation of the performance of the model on individual allergen categories, Figure A1, Figure A2, Figure A3, Figure A4, Figure A5 and Figure A6 in the appendix depict the percentage of false predictions for each allergen. Using the *Z*-score, we found no outliers (−3≤Z≤ 3) for the Color, Fine Lines, Folds, and Volumes modalities. On the other hand, we found PPD (Black Rubber Mix) to be an outlier for the Redness modality. Further investigation revealed this was caused by the black color of the substance appearing as high hemoglobin concentration in the modality. For Texture, the MCI/MI was found to be an outlier. Upon additional retrospective examination of this allergen’s images, MCI/MI was found to exhibit inconsistencies as far as the allergic reaction was concerned. More specifically, in our patient cohort the particular allergen frequently manifested two modes of action: erythema without textural aberrations, and textural aberrations without erythema. For that reason, MCI/MI requires further in depth investigation.

### 3.3. Third Experiment: Combined Modality Investigation

In our final experiment, we investigated the redundancy and the complementarity of the features the deep learning model learned between two different image modalities. To this end, we calculated and reported in Table 5 the mean and standard deviation of the Pearson correlation coefficient between all modality combinations on the validation data. We then employed a 5 × 2 cross-validation scheme and reported their *F*1 score for each modality in Table 6. We also performed an *F*-test between all possible modality pairs and the Color image modality, with the results presented in Table 7.

Our findings indicate that the correlations between different modalities vary, but they are all around 0.5 or higher. This means that there is a significant linear correlation between the learned features of the modalities. The modalities that exhibited the highest mean absolute Pearson coefficient were the Volumes + Folds, Texture + Folds, Texture + Fine Lines, as well as Fine Lines + Folds, implying that the network learned particularly redundant features for these pairs of modalities. By calculating the F1 score using a 5 × 2 cross-validation scheme, the best results were achieved by combining the features extracted from the Color and Redness modalities. This approach exhibited a statistically significant difference compared to using only the Color modality (p<0.05). However, combining either the Color or the Redness modality with other modalities did not yield better results than just using the Color modality. Additionally, the results were inferior for the feature combination involving only Volumes, Texture, Fine Lines, or Folds, that produced comparable results to the Color modality.

## 4. Discussion

In this study, we investigated the use of deep learning for diagnosing skin reactions in epicutaneous patch testing. Our methodology involves a multi-step process, starting with data collection from patients undergoing epicutaneous patch testing. Six photos were captured using the Antera 3D^®^ camera during each patient visit, in order to include all 30 allergens. To identify each individual allergen in the images, the clinician had to check the Fine Lines modality, captured 48 h after patch application, where the imprints of each chamber were visible on the skin texture. Using the camera’s bundled software, the allergens’ chamber borders were traced, and transferred to the 72 h images where they were not visible anymore. In the subsequent steps of our methodology, the core component is the deep learning model, a Convolutional Neural Network (CNN) which was trained to classify each allergen image between negative (no reaction or irritant reaction) and positive (allergic reaction). To ensure that the model received the most informative input, we employed an image pre-processing step which highlighted the allergen under examination while retaining contextual information. Our results indicated that the use of an alpha mask instead of cropping the input image, as well as a combination of the Color and Redness modalities as produced by the Antera 3D^®^ camera, are the most effective in binary classification.

We verified that cropping the input images to a specific ROI discards valuable contextual information and is therefore a sub-optimal pre-processing technique. To this end, we proposed the use of an alpha mask to highlight the ROI while retaining the overall image context. Furthermore, we extended all images to also include information from their adjacent patch areas. This method, which we labeled as CEAM, improved the *F*1 score over the cropped images, from 0.66 to 0.83. McNemar’s test further verified the statistical significance of the proposed pre-processing method. On the other hand, contrary to our intuition, we found no statistically significant difference between CEAM and AM, using McNemar’s test. This could be attributed to our limited data sample, and not having enough cases where the neighboring patch area affected an allergen to which a patient was negative to.

The results also suggest that textural information, contained in the Texture, Volumes, Fine Lines, and Folds modalities, are insufficient for classifying a reaction. Our analysis showed that Hemoglobin concentration, represented by the Redness modality, was the strongest predictor of a positive diagnosis, achieving a mean *F*1 score of 0.87 over 5-fold cross validation. In addition, our results showed that classification on plain Color images performed equally well as on the Redness modality. In particular, the *p*-value was found to be 0.760 when comparing the two, using McNemar’s test.

Individual allergen analysis revealed that the Redness image modality suffered from an abnormally high number of false predictions on the PPD (Black Rubber Mix). This result can be attributed to PPD’s color falsely appearing as high hemoglobin concentration in the redness image. For texture, MCI/MI were found to be outliers.

On the other hand, when combining different modalities, we found that the network learned complementary features between the Color and Redness image modalities, achieving greater results over just the Color image. Result significance was verified by the *p*-value of the statistical test (0.016). This was the only feature combination that we found added value in. Thus, we conclude that the advanced camera utilized can contribute to more accurate classification, but only when performing feature fusion through different modalities. The relatively high Pearson correlation coefficient calculated between the modality pairs indicates that overall the deep network learned features that were moderately linearly correlated, and therefore ensemble methods that combine different features could benefit from further optimization through feature selection and/or reduction techniques.

To the best of our knowledge, there is only one other work that deals with the problem of automated reaction diagnosis in epicutaneous patch testing. Chan et al. [10] collected a dataset of 3695 color images, each depicting individual allergens (similarly to our “Crop” pre-processing scheme). Only 118 of these images were labeled as positive. The authors used Google’s pre-trained Xception CNN, achieving an *F*1 score of 0.89, which is in agreement with our results.

### Limitations

This study has two limitations that need to be acknowledged.

Firstly, the sample size of our dataset was limited to 200 patients, which may hinder the generalizability of our results to larger populations. Additionally, our study focused on a specific set of allergens, presented in Table A1, and did not include other potential allergens that may be relevant in different countries, populations, or settings.

Secondly, our study utilized a single CNN architecture, namely EfficientNetB0. However, the performance of various CNN architectures on a similarly sized medical dataset has been investigated in previous work, and EfficientNetB0 was found to be one of the top performing networks [24].

## 5. Conclusions and Future Perspectives

In this study, we investigated the binary classification accuracy of a deep learning model for skin reaction detection using different pre-processing techniques and image modalities. We conducted three experiments, including a pre-processing scheme comparison, a modality performance comparison, and a multi-modality investigation. Our results showed that using a pre-processing scheme that retains contextual information can lead to improved classification accuracy, with the F1 score improving from 0.66 to 0.83. Regarding modality performance, we found the redness, a false color map of hemoglobin concentration, to produce the best results, reaching a mean recall of 0.86, specificity 0.94, and precision 0.89. However, we also found no statistically significant difference (*p* = 0.760) between using a color image and the redness, indicating that the two could be equivalent. On the other hand, combining these two image modalities did seem to provide additional value, as we observed a statistically significant difference (*p* = 0.016) between classifying a color image, and classifying a combination of the color and the redness image.

Overall, our study suggests that deep learning models have the potential to accurately detect and classify skin reactions in epicutaneous patch testing. However, it is important to note that our study has limitations, including the small sample size. Further studies with larger and more diverse datasets are needed to validate our findings and improve the generalizability of the model. Moreover, our study focused on binary classification, i.e., distinguishing between positive and negative patch test reactions. Future studies should explore the possibility of predicting severity as well. Additionally, while our study did not consider patient-specific clinical information such as atopic history, age, and gender, including such information in future studies could help to refine and individualize the diagnostic process.

By reducing the need for manual assessment, automated patch testing has the potential to greatly improve efficiency and reduce the workload of clinicians, allowing them to focus on more complex cases. Furthermore, automated patch testing can provide more objective and consistent results, minimizing the impact of observer bias.

## Figures and Tables

**Figure 1 bioengineering-10-00924-f001:**
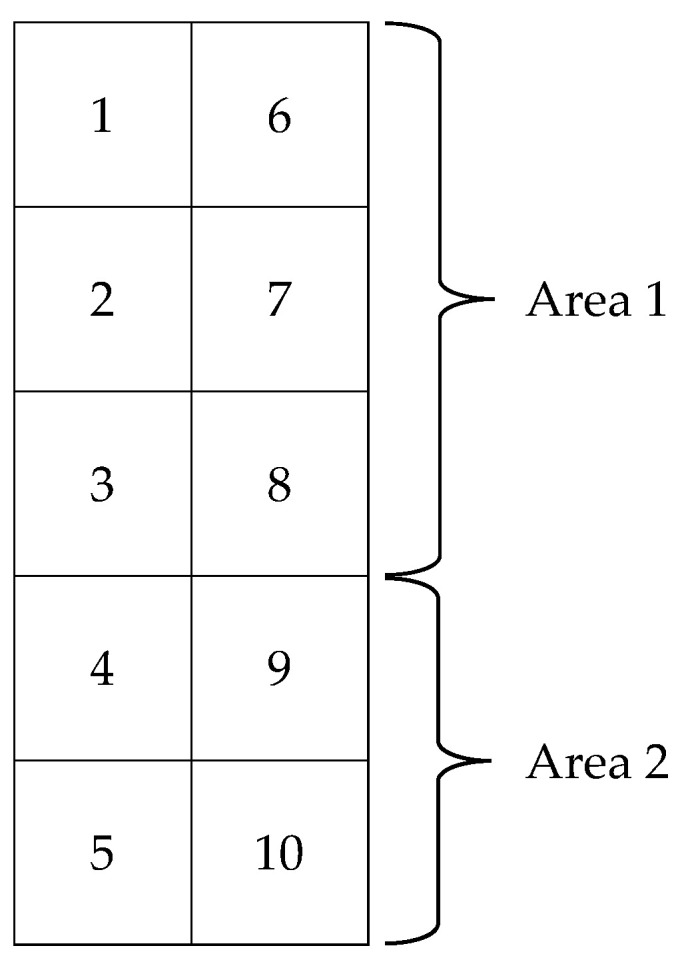
A single patch is broken down into two sub-areas, Area 1 and Area 2. Each area is captured independently with the Antera 3D^®^ camera due to its field of view being limited to 56 × 56 mm. Each number corresponds to a patch test chamber.

**Figure 2 bioengineering-10-00924-f002:**
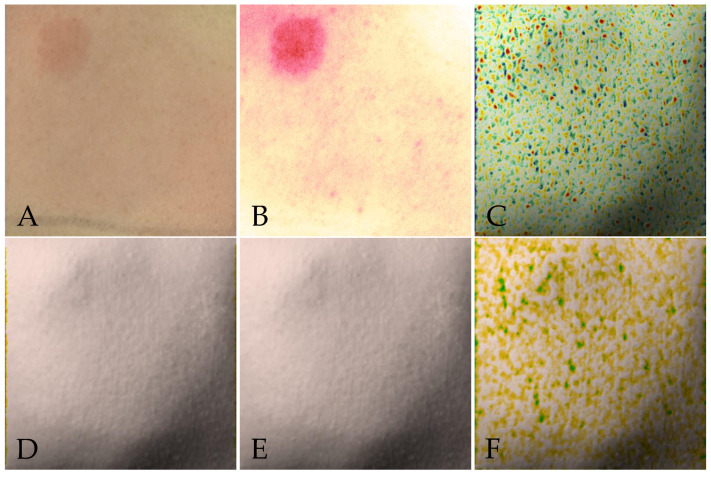
Samples of the different image modalities, for the same skin area of a patient. (**A**) Color; (**B**) Redness; (**C**) Texture; (**D**) Fine Lines; (**E**) Folds; (**F**) Volumes.

**Figure 3 bioengineering-10-00924-f003:**
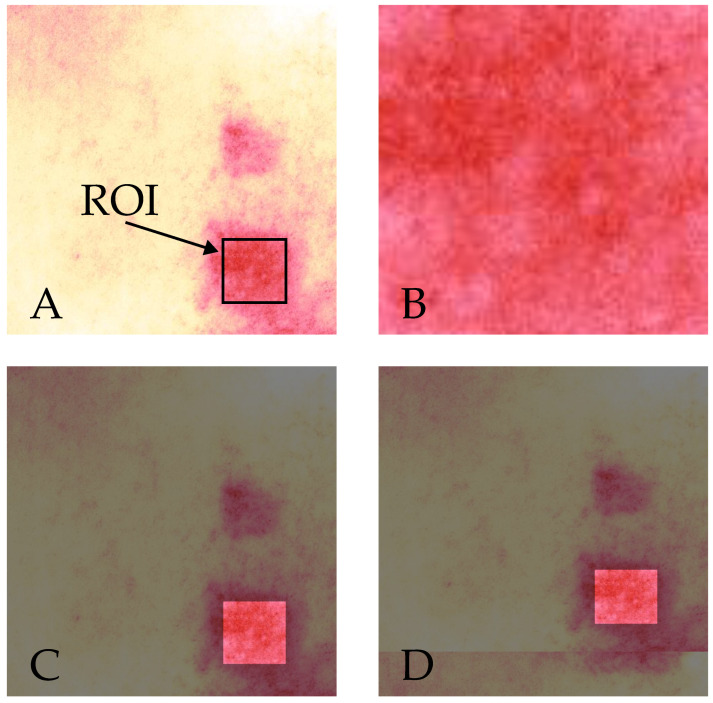
The result of the three image pre-processing schemes. (**A**) the original image, annotated to depict the Region of Interest (ROI); (**B**) the image is cropped to the ROI; (**C**) AM—the original image with an alpha mask applied; (**D**) CEAM—the image is alpha-masked with additional contextual information added from the adjacent patch test area.

**Figure 4 bioengineering-10-00924-f004:**
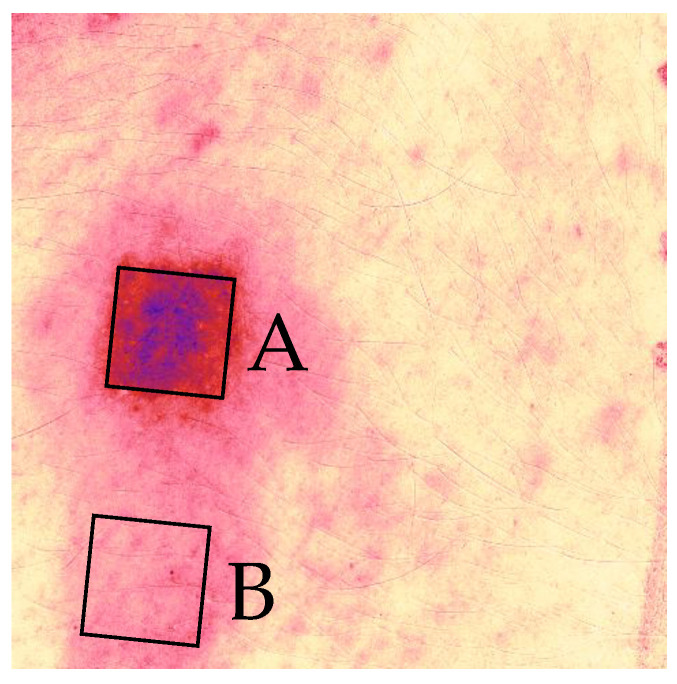
An example showcasing why contextual information is important. A strong reaction has manifested at allergen A. If allergen B is examined in isolation, it is likely to result in a false positive.

**Figure 5 bioengineering-10-00924-f005:**
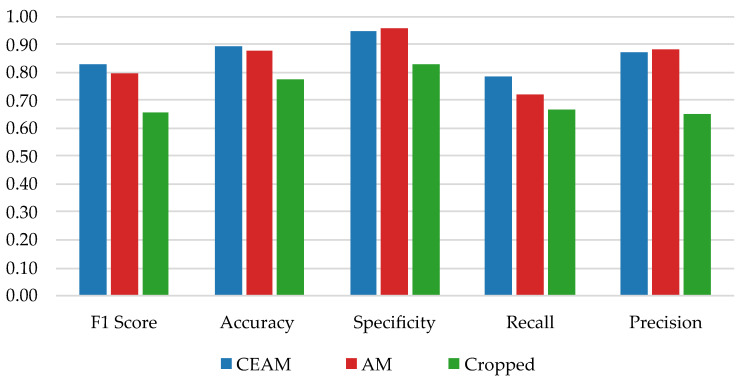
Performance metrics comparison for the three different pre-processing schemes.

**Figure 6 bioengineering-10-00924-f006:**
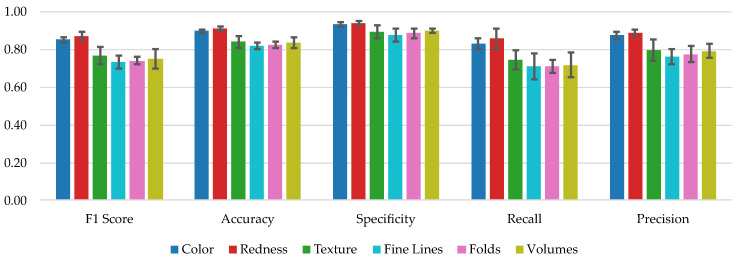
Performance metrics comparison for all the different image modalities. Each value is calculated as the mean across a 5-fold cross-validation. The error bar corresponds to the standard deviation.

**Table 1 bioengineering-10-00924-t001:** Performance metrics for each pre-processing scheme.

Pre-Processing Scheme	F1 Score	Accuracy	Specificity	Recall	Precision
CEAM	0.83	0.90	0.95	0.79	0.87
AM	0.79	0.88	0.96	0.72	0.89
Crop	0.66	0.78	0.83	0.67	0.65

**Table 2 bioengineering-10-00924-t002:** *p*-values corresponding to the pairwise comparison of different image pre-processing schemes, calculated using the McNemar test.

Hypothesis	*p*-Value
CEAM vs. AM	0.281
CEAM vs. Crop	**0.0**
AM vs. Crop	**0.0**

**Table 3 bioengineering-10-00924-t003:** Performance metrics for each modality.

Modality	F1 Score	Accuracy	Specificity	Recall	Precision
Color	0.85 ± 0.01	0.90 ± 0.01	0.94 ± 0.01	0.83 ± 0.03	0.88 ± 0.02
Redness	0.87 ± 0.02	0.91 ± 0.01	0.94 ± 0.01	0.86 ± 0.06	0.89 ± 0.02
Texture	0.77 ± 0.04	0.84 ± 0.03	0.89 ± 0.04	0.74 ± 0.05	0.79 ± 0.06
Fine Lines	0.73 ± 0.03	0.82 ± 0.02	0.88 ± 0.03	0.71 ± 0.07	0.76 ± 0.04
Folds	0.74 ± 0.02	0.83 ± 0.01	0.89 ± 0.03	0.71 ± 0.03	0.78 ± 0.04
Volumes	0.75 ± 0.05	0.84 ± 0.03	0.90 ± 0.01	0.72 ± 0.07	0.79 ± 0.04

**Table 4 bioengineering-10-00924-t004:** *p*-values corresponding to the pairwise comparison of different image modalities, calculated using the McNemar test (Table A3). The obtained *p*-values correspond to the additional 70/30 split and not the cross-validation results depicted in Figure 6.

Hypothesis	*p*-Value
Color vs. Redness	0.760
Color vs. Texture	**0.001**
Color vs. Volumes	**0.002**
Color vs. Folds	**0.000**
Color vs. Fine Lines	**0.001**
Redness vs. Texture	**0.000**
Redness vs. Volumes	**0.001**
Redness vs. Folds	**0.000**
Redness vs. Fine Lines	**0.000**
Texture vs. Volumes	0.890
Texture vs. Folds	0.061
Texture vs. Fine Lines	0.327
Volumes vs. Folds	**0.045**
Volumes vs. Fine Lines	0.255
Folds vs. Fine Lines	0.942

**Table 5 bioengineering-10-00924-t005:** Mean and standard deviation of the absolute Pearson coefficient values for each pair of modalities.

Modality Pair	Mean Correlation ± StdDev
Color + Redness	0.51 ± 0.07
Color + Volumes	0.51 ± 0.07
Color + Texture	0.52 ± 0.08
Color + Fine Lines	0.48 ± 0.06
Color + Folds	0.52 ± 0.07
Redness + Volumes	0.55 ± 0.08
Redness + Texture	0.59 ± 0.09
Redness + Fine Lines	0.55 ± 0.07
Redness + Folds	0.59 ± 0.07
Volumes + Texture	0.58 ± 0.08
Volumes + Fine Lines	0.58 ± 0.08
Volumes + Folds	0.62 ± 0.06
Texture + Fine Lines	0.60 ± 0.08
Texture + Folds	0.61 ± 0.08
Fine Lines + Folds	0.60 ± 0.08

**Table 6 bioengineering-10-00924-t006:** F1 score as calculated by the 5 × 2 cross-validation on each modality combination.

Modality	F1 Score
Color	0.70 ± 0.04
Color + Redness	0.74 ± 0.04
Color + Volumes	0.70 ± 0.05
Color + Texture	0.68 ± 0.05
Color + Fine Lines	0.69 ± 0.04
Color + Folds	0.69 ± 0.04
Redness + Volumes	0.68 ± 0.03
Redness + Texture	0.71 ± 0.02
Redness + Fine Lines	0.68 ± 0.03
Redness + Folds	0.69 ± 0.03
Volumes + Texture	0.56 ± 0.02
Volumes + Fine Lines	0.50 ± 0.03
Volumes + Folds	0.53 ± 0.03
Texture + Fine Lines	0.53 ± 0.04
Texture + Folds	0.52 ± 0.04
Fine Lines + Folds	0.47 ± 0.05

**Table 7 bioengineering-10-00924-t007:** *p*-values corresponding to the pairwise comparison of the classification of the features learned solely by the color modality, and the combination of features learned by pairing all other modalities, as calculated using the McNemar test.

Modality	*p*-Value
Color vs. Color + Redness	**0.016**
Color vs. Color + Volumes	0.373
Color vs. Color + Texture	0.353
Color vs. Color + Fine Lines	0.418
Color vs. Color + Folds	0.463
Color vs. Redness + Volumes	0.306
Color vs. Redness + Texture	0.583
Color vs. Redness + Fine Lines	0.430
Color vs. Redness + Folds	0.724
Color vs. Volumes + Texture	**0.014**
Color vs. Volumes + Fine Lines	**0.002**
Color vs. Volumes + Folds	**0.003**
Color vs. Texture + Fine Lines	**0.030**
Color vs. Texture + Folds	**0.010**
Color vs. Fine Lines + Folds	**0.007**

## Data Availability

Data available upon reasonable request.

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
