# Peer review of "Detecting Skin Reactions in Epicutaneous Patch Testing with Deep Learning: An Evaluation of Pre-Processing and Modality Performance"

_bioengineering, 2023, doi:10.3390/bioengineering10080924_

Round 1
Reviewer 1 Report
In this study, researchers explore the use of deep learning models and a dataset of multi-modal skin images to automate the identification of allergens causing Allergic Contact Dermatitis (ACD). They propose a deep learning approach with a context-retaining pre-processing technique and find promising results in combining color images and false-color maps of hemoglobin concentration to improve diagnostic accuracy. My comments are listed as follows:
- introduction: since the paper also addresses the pre-processing aspect, the introduction should provide an overview of the current methods for pre-processing dermoscopic images, ranging from heuristic approaches (doi: 10.1016/0016-0032(80)90058-7; doi: 10.2352/CIC.2004.12.1.art00008) to AI-based techniques (doi:10.1016/j.cmpb.2022.107040).
- Authors should provide additional background knowledge regarding the ICDRG criteria and explain why they are subjective.
- Page 6, Line 193: "To address this issue, we expanded each image under examination to incorporate pixels from its adjacent area. For images corresponding to patch area 1, we cropped 75 pixels from the corresponding patch area 2 and concatenated them at the bottom of the image." Please add the resulting image in Figure 3
- EfficientNetB0: why this network?
- The ICDRG criteria are designed to classify skin lesions into multiple classes based on dermoscopic features. However, in the specific context of the authors' study, they opt to work with binary classes. Why?
Author Response
In this study, researchers explore the use of deep learning models and a dataset of multi-modal skin images to automate the identification of allergens causing Allergic Contact Dermatitis (ACD). They propose a deep learning approach with a context-retaining pre-processing technique and find promising results in combining color images and false-color maps of hemoglobin concentration to improve diagnostic accuracy. My comments are listed as follows:
- Introduction: since the paper also addresses the pre-processing aspect, the introduction should provide an overview of the current methods for pre-processing dermoscopic images, ranging from heuristic approaches (doi: 10.1016/0016-0032(80)90058-7; doi: 10.2352/CIC.2004.12.1.art00008) to AI-based techniques (doi:10.1016/j.cmpb.2022.107040).
Response: We thank the reviewer for the information provided. We have expanded our “Skin analysis” section that mentions the camera’s capabilities of providing constant lighting conditions to acknowledge the importance of the matter and have cited the relevant work (citations 20, 21, 22). Please refer to pages 3 and 4, lines 128-135.
- Authors should provide additional background knowledge regarding the ICDRG criteria and explain why they are subjective.
Response: We thank the reviewer for the suggestion. We have revised the introduction accordingly to provide some background on the ICDRG criteria and cite the relevant source that mentions inter-observer variability as an issue in reading patch tests. Please refer to page 2, lines 37-48.
- Page 6, Line 193: "To address this issue, we expanded each image under examination to incorporate pixels from its adjacent area. For images corresponding to patch area 1, we cropped 75 pixels from the corresponding patch area 2 and concatenated them at the bottom of the image." Please add the resulting image in Figure 3
Response: Figure 3 was intended to only provide an example on how neighboring reactions can affect reading when cropping the image. However, the reviewer can refer to Figure 4 D, which depicts the resulting image requested.
- EfficientNetB0: why this network?
Response: This is a good question that was not adequately explained in our text. We have expanded section 2.2.1 to explain our reasoning behind this choice. Please refer to pages 5 and 6, lines 189-194. We also refer the reviewer to section 4.1, where we discuss the potential limitation by using only a single network.
- The ICDRG criteria are designed to classify skin lesions into multiple classes based on dermoscopic features. However, in the specific context of the authors' study, they opt to work with binary classes. Why?
Response: We thank the reviewer for their comment. This was also not explained in our manuscript, so we revised it accordingly. We have provided a breakdown with the percentages of each class in the dataset in section 2.1.5, which is then used to justify the choice of the binary classification problem in section 2.2. Please refer to page 5, lines 165-170, and lines 174-182.

Reviewer 2 Report
In this paper, authors collected a dataset of 1579 multi-modal skin images from 200 patients using the Antera 3D® camera. They investigated the feasibility of using a deep learning classifier for automating the identification of the allergens causing ACD. They also propose a deep learning approach that utilizes a context-retaining pre-processing technique to improve the accuracy of the classifier. The following comments are suggested to be considered:
1- The list of contributions should be clearly mentioned in the introduction section.
2- A paragraph describing the structure of the paper should be added at the end of the introduction section.
3- Confusion matrices should be added to clarify the recorded results.
4- The complexity of the proposed methodology should be discussed.
5- In the conclusion, the achieved results can be highlighted.
Author Response
In this paper, authors collected a dataset of 1579 multi-modal skin images from 200 patients using the Antera 3D® camera. They investigated the feasibility of using a deep learning classifier for automating the identification of the allergens causing ACD. They also propose a deep learning approach that utilizes a context-retaining pre-processing technique to improve the accuracy of the classifier. The following comments are suggested to be considered:
- The list of contributions should be clearly mentioned in the introduction section.
Response: We thank the reviewer for the suggestion. We have revised our introduction to include the list of contributions. Please refer to page 2, lines 63-69.
- A paragraph describing the structure of the paper should be added at the end of the introduction section.
Response: We have expanded the introduction with one additional paragraph describing the paper’s structure. Please refer to page 2, lines 75-80.
- Confusion matrices should be added to clarify the recorded results.
Response: We have added the confusion matrices for all modalities to Appendix B, and also cited them in Second Experiment’s text (section 3.2). Please refer to page 9 lines 326-327 and to table A2 in Appendix B, page 18.
- The complexity of the proposed methodology should be discussed.
Response: We have revised the Discussion section to summarize the proposed methodology and its complexity. Please refer to page 14, lines 372-385.
- In the conclusion, the achieved results can be highlighted.
Response: We have expanded the first paragraph of our conclusions to highlight our results. Please refer to page 15, lines 440-450 and page 16, lines 461-464.

Reviewer 3 Report
The authors presented a deep learning-based solution for epicutaneous patch testing. The experiments focus on investigating the influence of preprocessing methods and the use of different modalities. The authors demonstrated that combining two modalities yields promising results.
The paper is well-written, and the style is appropriate. The experimental design and methods address the research questions and the results are presented clearly. The conclusions drawn are supported by the obtained results.
There are minor issues that need to be fixed:
- Line 80: "Table A1 80 contains the complete list of the allergens used." - Please specify that the table is located in the appendix.
- line 97: "0h, 48h and 72h" Please ensure space is added between the numbers and the "h" symbol
Future Work: An interesting direction for future work would involve developing an end-to-end multimodal system capable of taking all modalities as input. This system would generate embeddings for each modality and aggregate them using mechanisms such as average pooling or attention pooling.
Author Response
The authors presented a deep learning-based solution for epicutaneous patch testing. The experiments focus on investigating the influence of preprocessing methods and the use of different modalities. The authors demonstrated that combining two modalities yields promising results.
The paper is well-written, and the style is appropriate. The experimental design and methods address the research questions and the results are presented clearly. The conclusions drawn are supported by the obtained results.
Response: We thank the reviewer for their positive comments.
There are minor issues that need to be fixed:
- Line 80: "Table A1 80 contains the complete list of the allergens used." - Please specify that the table is located in the appendix.
Response: We have revised the text accordingly. Please refer to page 3, lines 107-108.
- Line 97: "0h, 48h and 72h" Please ensure space is added between the numbers and the "h" symbol
Response: We have taken care of this formatting issue throughout our manuscript.
- Future Work: An interesting direction for future work would involve developing an end-to-end multimodal system capable of taking all modalities as input. This system would generate embeddings for each modality and aggregate them using mechanisms such as average pooling or attention pooling.
Response: We appreciate the reviewer's insightful suggestion for future work. This approach would enhance the integration of diverse modalities within a unified framework, facilitating a more holistic analysis of skin reactions. It is true that additional modalities, such as texture, should be complementary to the color and redness modalities, on which we plan to work in the future.

Reviewer 4 Report
The proposed study has limitations both in terms of its interest and the small sample size and diversity of cases. Consequently, the publication of this work brings no added value to the biomedical field.
English is generally correct
Author Response
The proposed study has limitations both in terms of its interest and the small sample size and diversity of cases.
Consequently, the publication of this work brings no added value to the biomedical field.
Response: We regret that the reviewer found our work of limited added value. However, it should be of note that the present work is the first in the literature reporting on the use of neural networks in atopic contact dermatitis and patch testing.
While the sample size of 200 patients may seem limited, for which the inclusion and exclusion criteria highlighted in the text as well as the advanced instrumentation required for the study should be taken into consideration, the reviewer should keep in mind that the total sample size is actually 1579 allergen images. Moreover, we have utilized statistical tests and cross-validation to ensure confidence in our conclusions.
Furthermore, the usage of the Antera 3D camera standardizes image acquisition, ensuring a homogenized dataset and keeping case diversity to a minimum.

Reviewer 5 Report
- How was the alpha value used in the alpha mask chosen?
- In line 203, it was mentioned that “only images obtained at 72h after exposure were used”. What’s the reason for this? Is it also true for other experiments?
- What’s the image modality used for experiment 1? Does the tendency (CEAM being the best) hold for different modalities?
- In experiment 1, the proposed method of using the original image with an alpha mask seems to include regions from other allergens. Are regions from other allergens considered contextual information? Please comment on this. Does it make sense to simply increase the size of the bounding box to include more contextual information instead of using the entire image?
- What’s the performance of the model on individual allergen categories? Are there any differences?
- In the third experiment, why only pairs of modalities were tested? Does it make sense to include all modalities for classification? What’s the redundancy in features from different modalities? Would some feature selection methods help the performance of using features from multiple modalities?
- Please comment on the false positive and false negatives samples from the deep learning models.
- Attention maps might be helpful for studying complementary information from different modalities.
- The contribution of this study to its field should be further illustrated. Is it applicable to the practice?
Author Response
- How was the alpha value used in the alpha mask chosen?
Response: This value was set arbitrarily so that there was a visible difference between the ROI and the rest of the image, as shown in fig. 4. We did not experiment with different alpha values. We have revised our text to clarify this. Please refer to page 6, lines 234-235.
- In line 203, it was mentioned that “only images obtained at 72h after exposure were used”. What’s the reason for this? Is it also true for other experiments?
Response: This is true for all experiments. As mentioned in the text in section 2.1.3, the maximum reaction was observed between day 3 and day 4 following allergen application. As such, we selected images at this timepoint as optimal for reading. We have updated our text to explain this more clearly by revising and moving this information to section 2.2.1. Please refer to page 6, lines 201-203.
- What’s the image modality used for experiment 1? Does the tendency (CEAM being the best) hold for different modalities?
Response: We thank the reviewer for catching this omittance. Experiment 1 was trained on the Color image modality, so we have edited lines 249-250 on page 7 to reflect this. We did not expand this experiment to additional modalities.
- In experiment 1, the proposed method of using the original image with an alpha mask seems to include regions from other allergens. Are regions from other allergens considered contextual information? Please comment on this. Does it make sense to simply increase the size of the bounding box to include more contextual information instead of using the entire image?
Response: Indeed, depending on the patch area (1 or 2) a single image may contain either 4 or 6 individual allergens. These are considered contextual information, as neighboring reactions can have a significant effect (figure 3 in our text). Centering the image on the allergen and increasing the bounding box around it does seem like a viable option, but one would first need to treat the bounding box size like a hyperparameter in order to find an optimal size. In addition, the allergens are usually located at the extremities of the images, meaning padding will almost definitely be necessary. As such, one might end up with an image where a large portion of it is just zeros. Therefore, we opted for the alpha mask as the simplest option that retains the maximum amount of information in the image.
- What’s the performance of the model on individual allergen categories? Are there any differences?
Response: We have provided a visual representation of the performance of the model on individual allergen categories for each modality in figures A1, A2, A3, A4, A5, A6 in Appendix C, pages 19-21. Additionally, we have computed the Z-score to find outliers, and commented on our findings. Please refer to page 12, lines 338-349.
- In the third experiment, why only pairs of modalities were tested? Does it make sense to include all modalities for classification? What’s the redundancy in features from different modalities? Would some feature selection methods help the performance of using features from multiple modalities?
Response: We appreciate the reviewer's insightful questions regarding the inclusion of all modalities and the redundancy in features when using multiple modalities for classification. We would like to address these points:
- We acknowledge that considering all modalities simultaneously for classification and training a new model to effectively combine them is an interesting direction for future research. However, the main issue we faced in this work is that we required a pre-trained model for fine-tuning our dataset on. These models are usually pre-trained on ImageNet, which expects a 3-dimensional RGB image as input. Therefore, attempts to concatenate the modalities resulted in reduced performance due to the different nature of the input data. Hence, in order to accurately compare the performance between a single modality and modality pairs, we ran cross validation on the test set’s extracted features. While we could combine more than two modalities, this would significantly increase computational requirements: 5 (cross-validation) times 26 possible combinations = 130 trainings. On top of this, aggressive feature selection would be required to address the thousands of features against a few hundred samples (as the cross validation is ran on the test set, therefore 30% of the data).
- To address the redundancy in features question, we have expanded our third experiment. We use the Pearson correlation coefficient to measure the strength of association that exists between all features of each modality combination. Please refer to page 9, lines 289-302, as well as page 12, lines 353-355, and lines 358-363.
- Please comment on the false positive and false negatives samples from the deep learning models.
Response: Overall, we did not find any allergens that were particularly predisposed to false negatives over false positives and vice-versa. We have not reported this numerically in the text as the samples were small for each allergen and the number of false positives or false negatives varied a lot. For example, Petrolatum was predominantly negative on patients, therefore the false positives outnumbered the false negatives. We refer the reviewer to the new graphs depicting misclassification percentages, as well as the computed confusion matrices over the 5-fold cross validation, which we reported in page 18, Appendix B, Table A2, which provide a more reliable overview of the classification performance.
- Attention maps might be helpful for studying complementary information from different modalities.
Response: We appreciate the reviewer's suggestion regarding the utilization of attention maps to study the complementary information from different modalities. However, in this study we employed EfficientNet, which does not incorporate an attention mechanism. Going forward, we acknowledge the importance of investigating attention mechanisms in the context of our proposed methodology.
- The contribution of this study to its field should be further illustrated. Is it applicable to the practice?
Response: This is the first study concerning the use of neural networks in contact dermatitis and patch testing. While the results are promising, there are important considerations for real-world implementation. Firstly, as we outline in our limitations section, further testing with larger and more diverse datasets is necessary to assess result generalizability. Furthermore, the image acquisition process needs to be streamlined to be used in daily clinical practice. We have revised our “Discussion” section to discuss on the proposed methodology complexity. Please refer to page 14, lines 372-385, and page 16, lines 461-463.

Round 2
Reviewer 1 Report
After reviewing the changes that were made, I believe that the manuscript has improved significantly. It is now much clearer and easier to follow, and the effort that was put into addressing the comments and suggestions provided is appreciated.
In particular, the addition of background knowledge has greatly enhanced the manuscript and made it accessible to a more general audience.
Reviewer 2 Report
Paper can be accepted in its current form.
Paper can be accepted in its current form.
Reviewer 5 Report
My comments have been addressed.